# The Interactive Effect of Elevated CO_2_ and Herbivores on the Nitrogen-Fixing Plant *Alnus incana* ssp. *rugosa*

**DOI:** 10.3390/plants10030440

**Published:** 2021-02-26

**Authors:** Haoran Chen, John Markham

**Affiliations:** Department of Biological Science, University of Manitoba, Winnipeg, MB R3T 2N2, Canada; john.markham@umanitoba.ca

**Keywords:** elevated CO_2_, nitrogen-fixing plants, herbivores, total phenolic compounds, C:N ratio

## Abstract

Many studies have found that future predicted CO_2_ levels can increase plant mass but dilute N content in leaves, impacting antiherbivore compounds. Nitrogen-fixing plants may balance their leaf C:N ratio under elevated CO_2_, counteracting this dilution effect. However, we know little of how plants respond to herbivores at the higher CO_2_ levels that occurred when nitrogen-fixing plants first evolved. We grew *Alnus incana* ssp. *rugosa* was grown at 400, 800, or 1600 ppm CO_2_ in soil collected from the field, inoculated with *Frankia* and exposed to herbivores (*Orgyia leucostigma*). Elevated CO_2_ increased nodulated plant biomass and stimulated the nitrogen fixation rate in the early growth stage. However, nitrogen-fixing plants were not able to balance their C:N ratio under elevated CO_2_ after growing for 19 weeks. When plants were grown at 400 and 1600 ppm CO_2,_ herbivores preferred to feed on leaves of nodulated plants. At 800 ppm CO_2_, nodulated plants accumulated more total phenolic compounds in response to herbivore damage than plants in the non-*Frankia* and non-herbivore treatments. Our results suggest that plant leaf defence, not leaf nutritional content, is the dominant driver of herbivory and nitrogen-fixing plants have limited ability to balance C:N ratios at elevated CO_2_ in natural soil.

## 1. Introduction

The growing conditions when major plant evolutionary events took place are quite different from the present. Vascular plants originated and adaptively radiated during the Early Silurian period *ca*. 440 million years ago [1]. The conditions during this period included an atmospheric CO_2_ level of 3300–3600 ppm [2]. The evolution of nitrogen-fixing plants (i.e., the symbiosis with nitrogen fixation bacteria involving the formation of root nodules) occurred during the rapid expansion of flowering plants in the late Cretaceous *ca*. 100 MYA [3]. During this time, atmospheric CO_2_ was *ca*. 1600 ppm, around four times the present atmospheric level [3,4,5,6]. The nitrogen-fixing symbiosis was likely more advantageous under conditions of ancient CO_2_ levels. Many studies have found that future predicted CO_2_ levels (<800 ppm) increases plant nitrogen fixation rates [7,8,9]. This increase has been shown to allow plants to maintain their leaf C:N ratio, whereas non nitrogen-fixing plants show increases in leaf C:N ratio with increasing CO_2_ levels [10]. However, these studies often use experimental protocols that provide plants with additional nutrients, especially P, which can stimulate an increase in nodule number or mass, but may not represent how nitrogen-fixing plants respond under natural soil conditions. Therefore, it is possible that the decrease in atmospheric CO_2_ over geological time restricted the success of existing, and evolution of new nitrogen-fixing species [6,11]. This would also explain the loss of the nitrogen-fixing trait that has occurred within the clade of nitrogen-fixing plants [11,12,13].

Many studies simulating predicted CO_2_ level increases in the coming decades have shown that small increases in atmospheric CO_2_ increase plant growth. Elevated atmospheric CO_2_ levels increase plant water use, allowing for increased carbon assimilation [14]. However, plant performance does not increase indefinitely with an increasing atmospheric CO_2_ concentration [15,16]. For example, it has been shown that elevating CO_2_ from 340 ppm enhanced non-nitrogen fixing plant seed yield by 30% to 40%, peaking at 1200 ppm CO_2_, and then decreasing as CO_2_ levels increase to 2400 ppm [15]. Increased carbon assimilation and storage of carbohydrates also have the effect of diluting other nutrients, such as nitrogen, within plant tissues [10,17,18]. This increase in leaf C:N ratio of tissues can alter plant interactions with other organisms and ecosystem processes.

There has been extensive work on plant interactions with herbivores at predicted future increases in CO_2_ levels on plants in general, with several studies including nitrogen-fixing plants [19,20]. While a higher leaf C:N ratio makes plant leaves less nutritious for herbivores, they may compensate by increasing their consumption rate to meet their growth requirements [17,21]. Elevated atmospheric CO_2_ can also increase plant carbon-based defensive compounds (e.g., phenolic compounds) and decrease nitrogen-based defensive compounds due to the dilution of N in leaves [17,22].

Since the ability of plants to form a nitrogen symbiosis can alter their response to CO_2_ levels, it may also alter their interaction with herbivores. Nitrogen-fixing plants can reduce plant tissue consumption by herbivores at elevated CO_2_ levels [20]. Nitrogen fixers can utilize C-based or N-based defensive compounds against herbivores [17,23] and the balancing of C:N ratio can affect the production of these defensive compounds. A study on nitrogen-fixing black alder showed leaf carbon-based (total phenols) and nitrogen-based (peroxidases and polyphenol oxidases) antiherbivore compounds increased in response to previous herbivore damage [23]. On the other hand, herbivores can strongly limit the abundance of some nitrogen-fixing plants and prevent their dominance in nitrogen-poor soils due to the herbivores’ preference for nitrogen-fixing plants with a high leaf nitrogen content [24]. Elevated CO_2_ (700 ppm) has been shown to shift herbivore preference to non nitrogen fixers (cotton), being consumed three times more than nitrogen fixers (alfalfa) [19]. This suggests that elevated CO_2_ can trigger a stronger effect on the herbivore-induced response of nitrogen fixers compared with non nitrogen fixers when plants are pre-damaged.

We hypothesize that when nitrogen-fixing plants first evolved in the late Cretaceous under high CO_2_ levels, they had a different relationship with herbivores compared with the present. There is no way of knowing how ancient plants functioned in the high CO_2_ levels they grew in. However, present day plant families diverged from one another when CO_2_ levels were much higher than at present, yet tend to respond in similar ways to small increases in CO_2_. This suggests that present day plants share traits with their ancient ancestors, making them suitable for understanding plant/herbivore interactions under ancient CO_2_ levels. No studies have examined the interaction between nitrogen-fixing plants and herbivores under ancient levels of CO_2_. The interaction between nitrogen-fixing plants and herbivores at a future level of CO_2_ may not reflect interactions in the Cretaceous since we know that plant performance peaks below the ancient level of CO_2_. The aim of this study was to look at the effects of future predicted (800 ppm) and Cretaceous period (1600 ppm) CO_2_ levels on nitrogen-fixing plant growth, leaf C:N ratio, and carbon and nitrogen based antiherbivore compounds. We compared nitrogen-fixing plants that formed nitrogen-fixing nodules to plants prevented from doing so. We also examined how growing plants at these CO_2_ levels affects preference for feeding on leaves of nitrogen-fixing or non-nitrogen fixing plants and the response of plants to feeding damage. We predicted that: 1. nitrogen-fixing plants can balance leaf C:N ratio under elevated CO_2_ compared to non-fixing plants, 2. as CO_2_ levels increase, carbon-based antiherbivore compounds will increase, and nitrogen-fixing plants will be able to maintain the levels of nitrogen-based antiherbivore compounds, 3. plants that accumulate more carbon-based or nitrogen-based antiherbivore compounds will have less herbivore damage, 4. herbivores prefer to feed on leaves from nitrogen-fixing plants when given a choice but, 5. herbivores consume more leaf tissue as the C:N ratio increases.

## 2. Results

### 2.1. Plant Biomass

Plants that developed nodules were 55 times larger than plants without (Figure 1). Nitrogen-fixing plants also increased biomass in response to increasing CO_2_, being *ca*. 1.5 larger at the 800 ppm and 1600 ppm CO_2_ level than plants grown at 400 ppm CO_2_. Non-nodulated plants showed no response to increasing CO_2_ (F = 51.84, *p* < 0.0001, for the inoculation treatment by CO_2_ treatment interaction). A non-linear (quadratic) model of nodulated plant biomass versus CO_2_ level predicted that plant biomass would peak at 1137 ppm CO_2_ (F = 39.50, *p* = 0.003, R^2^ = 0.63, Appendix A). Herbivory, by itself or crossed with CO_2_ level or *Frankia* inoculation, did not have a significant effect on plant total biomass.

### 2.2. Leaf C:N Ratio

*Frankia* inoculation decreased leaf C:N ratio, but only at the 400 ppm CO_2_ level (F = 3.41, *p* = 0.04 for the inoculation by CO_2_ level interaction, Figure 2). At 400 ppm, *Frankia* inoculated plants had a leaf C:N ratio that was half of that in the non-inoculated treatment. At 800 and 1600 ppm CO_2_, leaf C:N ratio was high, regardless of the plants being inoculated with *Frankia* or not.

### 2.3. Nitrogen Fixation

After plants had been fed on and artificially damaged, nitrogenase activity per plant was higher in plants grown at a higher CO_2_ level, regardless of whether or not they were part of the herbivory treatment (F = 14.73, *p* < 0.0001), increasing from 14.73 ± 1.01 µmol C_2_H_4_ h^−1^ plant^−1^ at 400 ppm CO_2_, to 20.59 ± 1.91 and 27.18 ± 1.41 µmol C_2_H_4_ h^−1^ plant^−1^ at 800 and 1600 ppm CO_2_, respectively (Appendix A). At the time of harvest, there was no effect of CO_2_ level (F = 0.84, *p* = 0.43) or herbivory (F = 0.00, *p* = 0.99) on rates of nitrogenase activity per plant, but the rate had dropped to 1.8 ± 0.1 µmol C_2_H_4_ h^−1^ plant^−1^ averaged over all treatments. When nitrogenase activity at the time of harvest was normalized to total plant leaf mass (F = 7.86, *p* = 0.0009), the rate decreased with increasing CO_2_ level (Figure 3), with no effect of herbivore damage (F = 0.28, *p* = 0.59) or interaction between herbivore damage and CO_2_ level (F = 0.81, *p* = 0.44). This was reflected in patterns of plant mass allocation to nodules, which decreased by 32%, from 2.20 ± 0.08% of total mass at 400 ppm CO_2_ to 1.60 ± 0.13 and 1.50 ± 0.07% at 800 and 1600 ppm CO_2_, respectively (F = 23.11, *p* < 0.0001), with no effect of herbivory (F = 1.97, *p* = 0.16), or interaction between CO_2_ level and herbivory (F = 1.5457, *p* = 0.2110). Stable isotope analysis indicated that all inoculated plants got most of their nitrogen from fixation, with a mean value of 98.56% ± 0.38%. Consequently, CO_2_ levels did not affect the proportion of nitrogen in the inoculated plants derived from fixation (F = 0.27, *p* = 0.76).

### 2.4. Leaf Damage and Antiherbivore Compounds

After five days of feeding, 18.24% ± 3.22% of the leaves on non-inoculated plants were damaged by the herbivores, compared to 12.97% ± 1.30% on the nodulated plants (F = 3.49, *p* = 0.06). Increased CO_2_ level had no effect on the proportion of leaves that were herbivore damaged (F = 1.48, *p* = 0.23).

Elevated CO_2_ increased leaf total phenolic concentration by the end of the growth period, increasing by 35% at 1600 ppm CO_2_ compared to 400 ppm and 800 ppm CO_2_ treatments (F = 27.06, *p* < 0.0001, Figure 4). Phenolic concentration also varied between herbivory treatments and plants with and without nitrogen-fixing nodules (F= 10.00, *p* = 0.003, for the interaction effect), but there was no three-way interaction between the treatments (F = 1.98, *p* = 0.15). We therefore examined the effect of *Frankia* inoculation and herbivore exposure at each CO_2_ level separately. At 800 ppm CO_2_, plants that were both able to fix nitrogen and exposed to herbivores had 18% higher phenolic concentration than nitrogen-fixing plants not exposed to herbivores or non-inoculated plants exposed to herbivores. At the other two CO_2_ levels, there was no effect of *Frankia* inoculation or herbivory exposure on leaf phenolic levels.

Specific polyphenol oxidase (PPO) activity was 84.2 ± 21.7 units min^−1^ mg^−1^ protein averaged across all treatments, and specific peroxidase (POD) activity was 11.0 ± 3.0 units min^−1^ mg^−1^ protein. The treatments had no effect on the level of this enzyme activity. There was also no relationship between antiherbivore enzyme activities and the degree of leave damage (F = 1.54, *p* = 0.22). We found no effect of the treatments on PPO activity when expressed on a leaf mass basis.

### 2.5. Herbivore Choice Experiment

The leaf consumption area was log transformed due to the large differences in variation between treatments. Averaged across all CO_2_ levels, herbivores preferred to feed on nitrogen-fixing plant leaves instead of leaves from non-nodulated plants (F = 74.14, *p* < 0.0001, for the inoculation effect). However, there was an interactive effect between CO_2_ level and *Frankia* inoculation on leaf consumption (F = 4.80, *p* = 0.01, Figure 5). Herbivores showed greater preference for leaves from inoculated plants that were grown at either 400 or 1600 ppm CO_2_, but there was no significant difference in the consumption of leaves between inoculated or non-inoculated plants grown at 800 ppm CO_2_. The greatest leaf consumption occurred on leaves from nitrogen-fixing plants raised at 1600 ppm CO_2_. There was no significant difference in the amount of leaf tissue consumed between non-inoculated plant growth at different CO_2_ levels. Pooling *Frankia* inoculated and non-inoculated plants together, herbivores consumed almost twice the mass leaf tissues grown at 1600 ppm CO_2_ compared with those grown at 800 ppm and 400 ppm (F = 3.25, *p* = 0.06, for a one-way ANOVA).

## 3. Discussion

Many plants benefit from small (industrial era level) increases in atmospheric CO_2_ by increasing their photosynthetic rate, which boosts growth and yield [25,26]. Without nodules, our plants performed extremely poorly in soil from the field where alders are usually found. Consequently, only nodulated plants showed a growth increase with an increase in atmospheric CO_2_. The lack of an increase in growth from 800 to 1600 ppm CO_2_, and predicted peak in growth at around 1100 ppm CO_2_, is consistent with other findings suggesting present day plants have a growth optimum below ancient levels of CO_2_ [16,27]. A meta-analysis also concluded that stimulation of plant growth by small CO_2_ increases is dependent on soil nutrient availability [28]. The authors reported that under high soil nitrogen, elevated CO_2_ increased aboveground plant growth by an average of 20.1%, while the response to elevated CO_2_ was only 8.4% under low soil nitrogen availability. Our study suggests that present day plants cannot acclimate the CO_2_ level under which symbiotic nitrogen fixation evolved, even when they have nitrogen-fixing nodules. Studies using small increases in CO_2_ have shown that nitrogen fixation rate was increased only when non nitrogen mineral nutrients were supplied [29]. While this suggests that nitrogen-fixing plants could respond positively to ancient CO_2_ levels if they are also supplied with these nutrients, it is unlikely that high nutrient availability was common in the past.

In our study, both future and ancient CO_2_ levels increased leaf C:N ratio, regardless of the ability of the plants to fix nitrogen. Therefore, the growth conditions in this experiment do not support our first prediction that nitrogen-fixing plants can balance their C:N ratio when CO_2_ levels increase to the levels we used. Previous studies have shown that non-nitrogen fixing plants leaf C:N ratio increased under elevated CO_2_ (720 ppm) when plants were grown in low (2.2 g N m^−2^ yr^−1^) or intermediate levels (6.7 g N m^−2^ yr^−1^) of nitrogen availability [30]. Increased C:N ratios due to high CO_2_ levels can lead to increased reactive oxygen species and leaf senescence [31,32]. The inability of nitrogen-fixing plants to maintain their C:N ratio that we found can be attributed to a reduced biomass allocation to nodules and, consequently, a decrease in the amount of nitrogen fixed per leaf mass under elevated CO_2_. There are contradictory reports of the effects of elevated CO_2_ on nitrogen fixation and plant nitrogen content [33]. Some studies show small increases in CO_2_ (550 ppm) stimulates symbiotic nitrogen fixation [34], with up to a doubling of the rate of nitrogen fixation in black locust (*Robinia pseudoacacia*) at 700 ppm CO_2_ [7]. In this latter study, plants were fertilized weekly with Hoagland-based nutrient solution for over one year, so other essential nutrients were not a limiting factor for nitrogen fixation. However, in natural ecosystems, nitrogen fixation may decrease in the long term with increasing CO_2_ due to the limitation of essential elements, especially phosphorus, molybdenum and iron [29,35]. Hungate et al. [36] found that nitrogen fixation at elevated CO_2_ in *Galactia elliottii* Nutt. in scrub-oak vegetation in central coastal Florida increased during the first year but then declined by the third year and subsequent years. This was due to Molybdenum, a required cofactor for nitrogenase, becoming limited under elevated CO_2_. Edwards et al. [37] also found that elevated CO_2_ (700 ppm) had no effect on nitrogen fixation in white clover (*Trifolium repens*) with a low phosphate application (0.04 mM), but increased with a high phosphate application (1.0 mM). The meta-analysis of de Graff et al. [28] found that elevated CO_2_ increased nitrogen fixation by 51% when nutrients other than nitrogen were also applied, but had no effect on nitrogen fixation in the absence of nutrient additions. Our study found that whole-plant nitrogen fixation rate increased under elevated CO_2_ at week 14–15, but dropped by week 19 regardless of CO_2_ levels. As our whole plant rates of nitrogen fixation are a function of plant mass, the early differences in fixation may simply be due to differences in plant mass. The decrease in nitrogen fixation over time suggested increasing nutrient limitation on this process, which is to be expected since our fertilization stopped after 12 weeks. It is possible that the nutrient-poor soil we used prevented nodulated plants from being able to maintain their tissue C:N ratio. It is also likely that in the future, nitrogen-fixing plants will find themselves in conditions with less available nutrients as the increased carbon content of litter from rising CO_2_ levels can result in immobilization of plant nutrients [38].

We did find evidence for our second prediction. Carbon-based antiherbivore compounds did increase with increasing CO_2_. Additionally, nitrogen-fixing plants did maintain their levels of N-based antiherbivore compounds, even though the nitrogen-fixing plants were not able to maintain their C:N ratio. We also found that having the ability to fix nitrogen increased total phenols when the plants were exposed to herbivores at 800 ppm CO_2_. This supports other studies showing that nitrogen fixation increases inducible herbivore defence [23,39] and may provide some feeding deterrence at future, but not ancient, CO_2_ levels.

Our results also confirmed the third prediction that plants get less leaf damage when leaf tissue accumulates more total phenolic compounds at elevated CO_2_. However, we did not find that the nitrogen-based antiherbivore compound production was related to the degree of plant damage from herbivores. Total phenolic compounds play a major role in host resistance to herbivores [40,41,42] because they can bind to insects’ digestive enzymes and interfere with animal digestion [43]. This defence mechanism can be present constitutively or induced after damage by herbivores [39]. Several studies have shown that plants exhibit many inducible defence mechanisms [23,39]. They can both deter feeding and retard insect development [44]. Previous studies have also shown that total phenolic compounds of maize (*Zea mays*) increased under elevated CO_2_ [45]. Previous studies have also shown that plants can accumulate more N-based secondary compounds (e.g., polyphenol oxidase, peroxidase) to defend against herbivore damage [39,46]. These compounds catalyze the oxidation of phenolics to quinones, which bind to leaf protein and inhibit protein digestion in herbivores [39,46]. Our results support the previous study that *Alnus* exhibits an inducible defence mechanism in response to herbivore feeding by maintaining polyphenol oxidase and peroxidase activity under elevated CO_2_.

The choice experiment showed that herbivores generally prefer to eat the nodulated, rather than non-nodulated plants, confirming our 4th prediction. Herbivores also increased consumption at higher CO_2_, confirming our 5th prediction that herbivores consume more when food quality goes down. However, there was no difference in herbivore preference between nodulated and non-nodulated plant leaves when grown at 800 ppm CO_2_. This may be related to the fact that at this CO_2_ concentration, nitrogen-fixing plants had a higher concentration of phenolic compounds when they had previously been fed on, suggesting they can have a greater induced antiherbivore response. This result also supports a previous study suggesting that plant leaf defence, not leaf nutritional content, is the dominant driver of herbivore preference [47].

## 4. Materials and Methods

### 4.1. Plant Growth Condition and Treatments

Seedlings of speckled alder (*Alnus incana* ssp. *rugosa*) were raised in growth chambers with 16-h day time/8-h night time (22 °C/18 °C, day/night). The plants were grown in a mix of field soil from a *Pinus banksiana* forest stand that contains *Alnus viridis* ssp. *crispa*, mixed with an equal volume of Turface. A previous study has shown that the soil from the site has an inorganic nitrogen level of 10.2 ± 0.6 mg/kg and an extractable phosphate level of 0.98 ± 0.33 mg/kg [48]. Plants were fertilized with 1/16 N-Rorison nutrient solution containing 0.125 mM Ca(NO_3_)_2_, 1 mM CaCl_2_, 1 mM K_2_HPO_4_, 1 mM MgSO_4_, 0.0534 mM Fe-EDTA, 0.009 mM MnSO_4_, 0.0045 mM H_3_BO_3_, 0.001 mM Na_2_MoO_4_, 0.0015 mM ZnSO_4_, and 0.0015 mM CuSO_4_ [49] once a week for the first 12 weeks. The fertilization was stopped before the choice experiment, described below. Half of the plants were inoculated with the nitrogen-fixing symbiont *Frankia*, which came from crushed root nodules collected from wild plants. The non-inoculated plants received the same dose of inoculum, which was first autoclaved. Plants were divided among six growth chambers, with two chambers each receiving one of three atmospheric CO_2_ levels: 400, 800, and 1600 ppm, representing ambient, future and Cretaceous era atmospheric levels, respectively, following Murray et al. [50]. There were 30 plants per combination of inoculation treatment and CO_2_ level. In each chamber, carbon dioxide levels were continuously monitored and controlled with a CO_2_ sensor connected to a microcontroller, which controlled a CO_2_ injection system [51].

### 4.2. Herbivore Choice Experiment

At week 13 after the start of the CO_2_ treatments, a choice experiment was set up to determine whether insects prefer non-nodulated or nodulated plants grown at each CO_2_ level. We randomly paired one leaf from an inoculated plant with one leaf from a non-inoculated plant grown at the same CO_2_ level. Each pair of leaves were placed into a Petri dish with moistened filter paper and along with one white-marked tussock caterpillar (*Orgyia leucostigma*) larva. The larvae were raised from eggs obtained from the Great Lakes Forestry Centre (Natural Resources Canada) on their artificial diet at 22 °C on a 12 h light: 12 h dark cycle for one month in room level (*ca*. 400 ppm) CO_2_. Insects were starved for 48 h before placing them into the Petri dishes. Petri dishes were placed under the same conditions where larvae were raised. Nine inoculated and nine non-inoculated leaves were collected from each chamber, for a total of 18 replicates at each level of CO_2_. Leaf areas were measured before and after 24 h of insect feeding to determine insect preference.

### 4.3. Herbivores Treatment

After the choice experiment, half of the plants in each combination of inoculation treatment and CO_2_ level (15 plants) were randomly assigned to an herbivore exposure treatment of *O. leucostigma*, raised and starved as above. The rest were non-herbivore controls. Since it was not possible to control insect movement, all plants exposed to herbivores were moved to one chamber at each CO_2_ level for the feeding period and then moved back after feeding. Each plant was exposed to an average of 5 insects (for a total of 145 larvae in each chamber) for five days without controlling the insect’s movement. After the insects were removed, the proportion of leaves showing signs of insect feeding was assessed. Then, half of each plant’s leaves were artificially removed to standardize plant damage. The plants were grown for another six weeks after artificial leaf removal and then harvested.

### 4.4. Nitrogen Fixation

The activity of the nitrogen-fixing enzyme, nitrogenase, was measured after leaves were damaged (at week 14–15) and at the time of the harvest (week 20) using acetylene reduction assays [49]. For the pre-harvest assays, the acetylene reduction rate was expressed per plant. At the time of harvest, the acetylene reduction rate was also calculated per total mass and leaf dry mass, to give an index of the amount of nitrogen fixed per mass of photosynthetic tissue. After harvest, dry plant leaf tissues were ground in a ball mill. Leaf carbon and nitrogen content, and their isotopic ratios were measured at the Stable Isotope Facility at the University of California, Davis. The percentage of nitrogen derived from fixation (%Ndfa) was calculated using a modification of the equation following Boddey et al. [52]:$$\%\mathrm{Ndfa}\;=\frac{\left(\delta^{15}N_{reference\;}-\;\delta^{15}N_{fixing\;plant}\right)}{\left(\delta^{15}N_{reference}\;-\;B\right)}\times 100$$where *δ*^15^*N_reference_* is the level of *δ*^15^*N* in the uninoculated plants, *δ*^15^*N_fixing plant_* is the level of *δ*^15^*N* in the inoculated plants (*Alnus incana* ssp. *rugosa*) and B is the *δ*^15^*N* of nitrogen fixing plants that are fully dependent upon symbiotic nitrogen fixation and sampled at the same growth stage.

### 4.5. C-Based Antiherbivore Compounds

The C-based antiherbivore compounds were analyzed at the time of harvest. Total phenolic compounds were determined using the Folin–Ciocalteu method [53]. In brief, 0.01 g dry leaf powder was incubated in the dark for 24 h in 10 mL of 40% ethanol. After centrifuging at 5000 RPM for 10 min, 1 mL of supernatant was mixed with 0.5 mL 50% Folin–Ciocalteu reagent, incubated for 3 min and then incubated in the dark for 30 min with 1 mL of 5% Na_2_CO_3_. The absorbance was then measured at 750 nm and compared to a standard curve prepared with known concentrations of gallic acid.

### 4.6. N-Based Antiherbivore Compounds

The N-based antiherbivore compounds were analyzed after leaf damage. The polyphenol oxidase and guaiacol peroxidase assays were only performed on *Frankia* inoculated plants since the non-inoculated plants did not produce enough tissue to conduct all assays. Polyphenol oxidase and guaiacol peroxidase activity were measured relative to the protein content of leaves. Leaf protein content was determined with a Bio-Rad protein assay using the Bradford [54] method. In brief, 0.6 g of frozen leaf tissue was ground with liquid nitrogen and then homogenized in 10 mL extraction buffer consisting of 50 mM K-phosphate (pH 7.1), 1% PVP (polyvinyl-pyrrolidone), 1 mM EDTA (Ethylenediaminetetraacetic acid), and 5 mM ascorbic acid. The homogenate was centrifuged at 23,000× *g* for 20 min at 4 °C. The supernatant was used for both the protein and antiherbivore compound assays. The protein content was measured after adding the Bradford reagent, using BSA as a standard. Polyphenol oxidase and guaiacol peroxidase were measured following Tscharntke et al. [23]. For the polyphenol oxidase assay, 0.15 mL of supernatant was mixed with 1.1 mL of 50 mM potassium phosphate (pH 7.1) and 0.3 mL of 100 mM catechol. After two minutes, the absorbance at 420 was measured. One unit of polyphenol oxidase was defined as the amount of enzyme that caused an increase in the absorbance of 0.01 per minute, and specific activity was expressed as units min^−1^ mg protein^−1^ [55]. The guaiacol peroxidase assay was performed by mixing 0.15 mL of supernatant with 1.2 mL of 50 mM K-phosphate (pH 7.1), 0.15 mL of 20 mM guaiacol, and 0.06 mL of 12.3 mM H_2_O_2_. The enzyme activity was calculated from the increase in the absorbance at 470 nm after 2 min [23]. One unit of activity was defined as the amount of enzyme that increases the optical density 470 by 0.01 per min. The specific enzyme activity was expressed as a change in optical density (ΔOD) per min per mg protein (units min^−1^ mg^−1^ protein).

### 4.7. Statistical Analysis

Data were analyzed using JMP Pro 14. The data were analyzed using three-factor (CO_2_ level, *Frankia* inoculation, and herbivore exposure) ANOVA models followed by Tukey’s HSD tests, with growth chamber nested within CO_2_ treatments. The non-inoculation treatment was dropped from the model for the nitrogen fixation response variables, as non-inoculated plants did not fix nitrogen. We also explored the relationships between variables using linear and non-linear least squares models. Since the effect of the CO_2_ level on a number of plant responses was not linear, quadratic models were also used to fit the data. Residual plots were used to determine if there are large differences in variation between treatments. If so, the data were log-transformed.

## 5. Conclusions

Our study found that nitrogen fixation cannot meet the higher nitrogen demand of plants when grown under 800 and 1600 ppm CO_2_, the former being a CO_2_ concentration that may occur in the future and the latter a level that occurred in the distant past. The increased leaf C:N ratio under elevated CO_2_ is associated with higher total leaf phenol content. Like non-nitrogen fixing plants, nitrogen-fixing plants at present have likely evolved not to be able to acclimate ancient CO_2_ levels. Future studies are needed to test the effect of ancient CO_2_ on nitrogen fixation when other nutrients are not limited. Finally, field studies are required to examine the complex relationship between elevated CO_2_, nitrogen fixation and herbivory.

## Figures and Tables

**Figure 1 plants-10-00440-f001:**
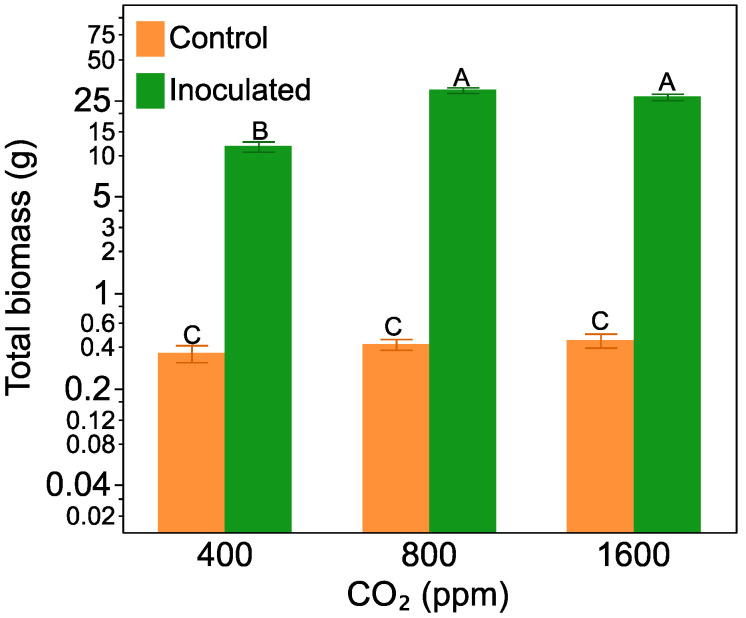
The effect of CO_2_ level on *Alnus incana* ssp. *rugosa* total biomass with and without nitrogen-fixing nodules. Bars are means with standard errors and averaged across non-herbivore and herbivore treatments. Bars with the same letters are not significantly different according to Tukey’s HSD Test.

**Figure 2 plants-10-00440-f002:**
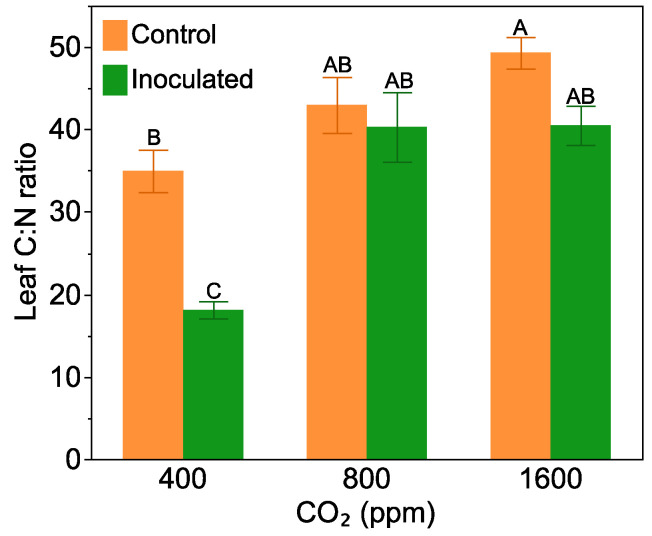
The effect of *Frankia* inoculation and CO_2_ level on leaf C:N ratio. Bars are means with standard errors. Bars with the same letters are not significantly different according to Tukey’s HSD Test.

**Figure 3 plants-10-00440-f003:**
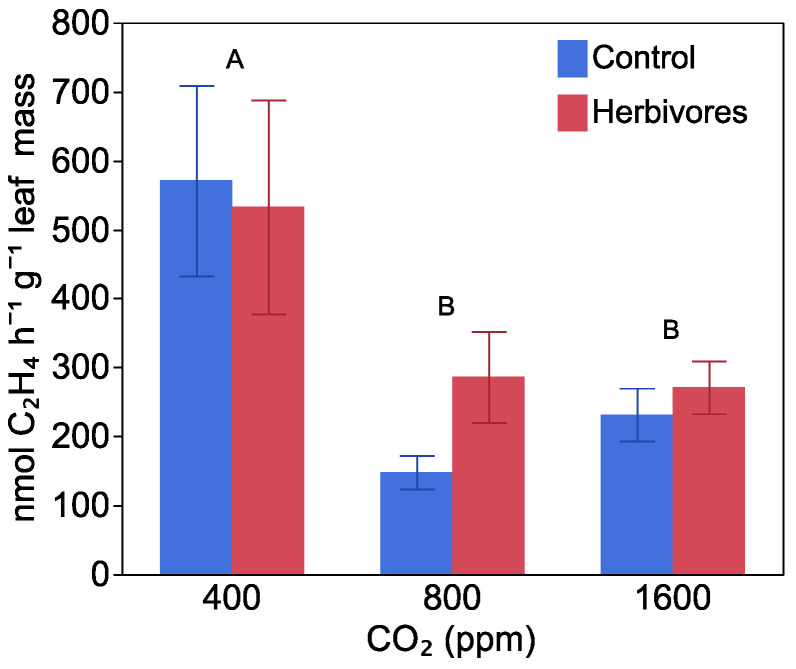
The effect of CO_2_ level and herbivores on nitrogenase acivity per leaf mass at harvest. Blue bars indicated plants without herbivore damage, and red bars indicated with herbivore damage. Different letters indicate significant differences between CO_2_ levels. There was no difference between herbivore and non-herbivore treatments. Bars are means with standard error.

**Figure 4 plants-10-00440-f004:**
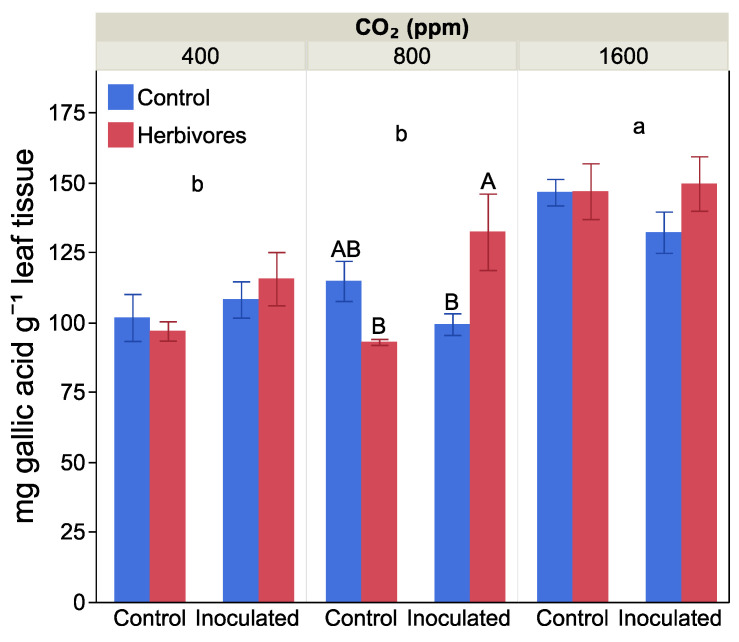
The effect of CO_2_ level, *Frankia* inoculation and herbivore exposure on leaf total phenolic compounds. Different small letters indicate significant differences among the three levels of CO_2_, averaged by *Frankia* and herbivore treatments. The different capital letters indicate significant differences between *Frankia* and herbivore exposure treatments within a CO_2_ level. Bars are means with standard error.

**Figure 5 plants-10-00440-f005:**
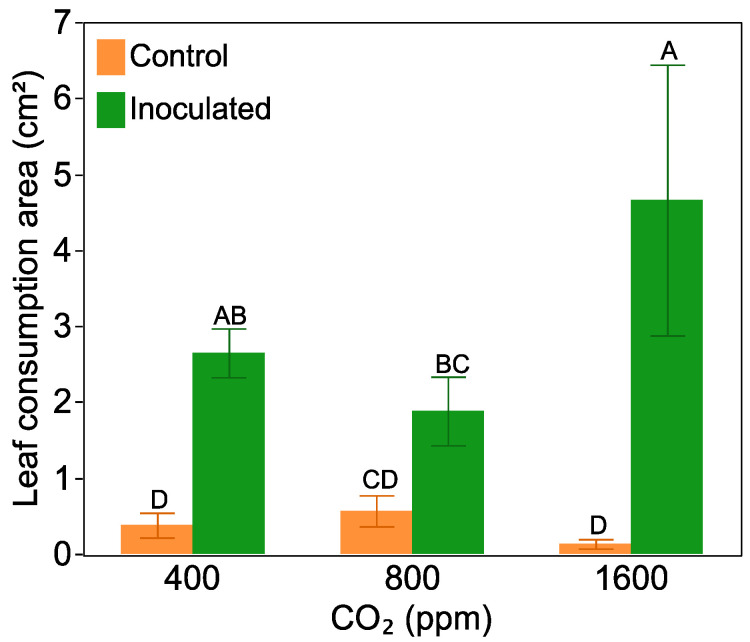
Insects consumption area on non-inoculated and inoculated plant leaves for one day at three levels of CO_2_ (400 ppm, 800 ppm, 1600 ppm) at week 13. Bars with the same letters are not significantly different according to a Tukey’s HSD test. Bars are mean value with standard error.

## Data Availability

Data are available from the corresponding author.

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
