# Peer review of "The Interactive Effect of Elevated CO_2_ and Herbivores on the Nitrogen-Fixing Plant *Alnus incana* ssp. *rugosa"

_plants, 2021, doi:10.3390/plants10030440_

Round 1

Reviewer 1 Report

Review of the article The Interactive Effect of Elevated CO2 and Nitrogen-Fixing Plant.
In the current version, the authors evaluated whether increasing CO2 levels will change the C:N ratios and whether this increase will change the N2 fixation capacity of Alnus incana and its susceptibility to herbivores attack. The authors present antecedents in the introduction in order to follow the research questions which are:
 1. Nitrogen-fixing plants can balance leaf C:N ratio under elevated CO2 compared to non-fixing plants
 2. as CO2 levels increase, carbon-based antiherbivore compounds will increase, and nitrogen-fixing plants will be able to maintain the levels of nitrogen-based antiherbivore compounds
3. plants that accumulate more carbon-based or nitrogen-based antiherbivore compounds will have less herbivore damage
4. herbivores prefer to feed on leaves from nitrogen-fixing plants when given a choice but
5. herbivores consume more leaf tissue as the C:N ratio increases.
I think, that these several questions can be summarized in a single sentence, in fact, questions number 2 and 3 are similar, the same is applicable for questions 4 and 5 although they are not the same, both are complementary.
The authors designed an experiment with inoculated and not inoculated plants of Alnus incana growing at three different CO2 levels (400, 800 and 1600 ppm) in order to measure N2 fixation, plant biomass, C and N content (C:N ratio), phenolics compounds and herbivore preference. The experiment is well designed and the statistical procedures are standards.
The main results showed that CO2 levels did not affect the biomass of non-inoculated plants while the increase from 400 to 800 affected positively the biomass of N2-fixing plants (no difference was observed for 800 and 1600 ppm of CO2). There was a huge difference in biomass content for non-inoculated and N2 fixing plants. It would be very nice if the authors explain why they did not include N fertilizer treatments as control. Leaf C:N ratio increased as CO2 level increased as predicted by the authors although the N2 fixation decreased and the herbivore presence did not affect this parameter. Herbivores prefer to feed leaves from fixing plants growing at high CO2 levels, although the data given had high variation. Total phenolic compounds showed a trend to increase as CO2 increased but the data are not all significant.
The conclusions must be based on their own results (line 402: Nutrient limitation likely prevents the nitrogen fixation rate increasing under elevated CO2), not on data that were not measured.
I would like to suggest to the authors to change the title since this study is only limited to Alnus incana and can not be extended to other species.
Minor spelling mistakes (lines 218 and 243).
Format: I would like to suggest the authors reduce the distance from Y-labels to the axis, X label below the axis. In figure 1, the same size for the numbers in Y-axis. Bar labels inside the graph.

Best regards

Reviewer 2 Report

The paper submitted by Chen and Markham describes experiments to inform several hypotheses around the effect of elevated CO2 on biomass production by alder, nitrogen fixation and attraction to herbivores (caterpillars of O. leucostigma). The manuscript is well written but there are some sporadic minor grammatical /typographical errors.

While the results support four of the five hypotheses, there is a flaw in the overall hypothesis stated in line 74. While the question posed is fascinating, this reviewer cannot see how it could be tested without access to germline as it was during the Cretaceous. Surely alder found under present day conditions have adapted to changing CO2, and their response to CO2 levels of the “distant past” cannot be assumed to be as it was then. The same holds for behavior of herbivorous insects in response to plant phenolics etc. The authors are, therefore, requested to revise the stated hypotheses as they relate to “prehistoric CO2 levels”.

Specific comments:

  1. Lines 15-16: “…after grown for 19 weeks…” – please correct.
  2. Line 36: “These studies often provide plants with additional …” – I assume that experiments described in the studies entailed inclusion of additional nutrients.
  3. Line 68: “…. Limit some nitrogen fixer’s abundance and…” abundance of the plants or the bacteria?
  4. Line 120: “After plants were fed on _____ and one week after…” Something is missing here.
  5. Line 164: “CO2(ppm)” appears out of place here.
  6. Line 342: “was also calculate per total…” – should read “calculated”.
  7. Line 342: Please include details on the acetylene reduction assay, or if you followed a previously published procedure, just refer to that.
  8. Line 398: “..when growth under…” – should read “grown”
  9. Supplementary Fig S3 legend: “…in my previous study…” – rather refer to the reference if this was published, or alternatively to the thesis / dissertation.
